# Requirement for highly efficient pre-mRNA splicing during *Drosophila* early embryonic development

**Leonardo Gastón Guilgur[1,2,3], Pedro Prudêncio[1,2,3], Daniel Sobral[1], Denisa Liszekova[1], André Rosa[1], Rui Gonçalo Martinho[1,2,3]\***

[1]Instituto Gulbenkian de Ciência, Oeiras, Portugal; [2]Departamento de Ciências Biomédicas e Medicina, Universidade do Algarve, Faro, Portugal; [3]IBB-Institute for Biotechnology and Bioengineering, Centro de Biomedicina Molecular e Estrutural, Universidade do Algarve, Faro, Portugal

**Abstract** *Drosophila* syncytial nuclear divisions limit transcription unit size of early zygotic genes. As mitosis inhibits not only transcription, but also pre-mRNA splicing, we reasoned that constraints on splicing were likely to exist in the early embryo, being splicing avoidance a possible explanation why most early zygotic genes are intronless. We isolated two mutant alleles for a subunit of the NTC/Prp19 complexes, which specifically impaired pre-mRNA splicing of early zygotic but not maternally encoded transcripts. We hypothesized that the requirements for pre-mRNA splicing efficiency were likely to vary during development. Ectopic maternal expression of an early zygotic pre-mRNA was sufficient to suppress its splicing defects in the mutant background. Furthermore, a small early zygotic transcript with multiple introns was poorly spliced in wild-type embryos. Our findings demonstrate for the first time the existence of a developmental pre-requisite for highly efficient splicing during *Drosophila* early embryonic development and suggest in highly proliferative tissues a need for coordination between cell cycle and gene architecture to ensure correct gene expression and avoid abnormally processed transcripts.

**\*For correspondence:**
rmartinho@igc.gulbenkian.pt

**Competing interests:** The authors declare that no competing interests exist.

**Reviewing editor**: Elisa Izaurralde, Max Planck Institute Development Biology, Germany

## Introduction

Timing and coordination of biological processes is crucial for cellular homeostasis and normal development. *Drosophila melanogaster* embryonic development starts with thirteen nuclear divisions without cytokinesis (syncytial blastoderm), these divisions being among the fastest known for any animal embryonic system (*Foe and Alberts, 1983*). *Drosophila* syncytial blastoderm formation relies on the maternally encoded gene products loaded into the egg during oogenesis (*Tadros and Lipshitz, 2009*). After fertilization, as nuclei enter interphase 14, the maternal to zygotic transition (MZT) occurs in which the soma suddenly becomes transcriptionally active and many of the maternally encoded gene products are rapidly degraded (*Anderson and Lengyel, 1979*; *McKnight and Miller, 1976*; *Yasuda et al., 1991*). While the major burst of zygotic transcription occurs once the nuclei arrest in interphase 14, there is an initial wave of zygotic gene expression during the syncytial nuclear divisions 8–13 (*Pritchard and Schubiger, 1996*; *ten Bosch et al., 2006*). Due to the extreme speed of syncytial nuclear divisions, a limitation to the size of early zygotic transcriptional units has been suggested (*McKnight and Miller, 1976*; *Rothe et al., 1992*; *Shermoen and O'Farrell, 1991*). Consistently, approximately 70% of early zygotic genes are small in size and intronless (*De Renzis et al., 2007*). As only 20% of *Drosophila* genes are intronless, it has been proposed that small intronless genes have an important selective advantage for transcription during the syncytial blastoderm formation (*De Renzis et al., 2007*).

**eLife digest** When a fertilized egg develops into an embryo, the expression of many genes must be carefully timed and coordinated. Researchers regularly use a type of fruit fly called *Drosophila* to study development because it is small, it has a short lifespan, and its whole genome sequence is already known. The development of a *Drosophila* embryo starts with the nucleus of the fertilized egg, which contains most of the cell's genetic material, dividing 13 times in quick succession, without the cell itself splitting. These divisions are amongst the fastest known for any animal, and given the fast developmental speed, the embryo must efficiently express all genes it needs to stay alive. Because cell division is known to inhibit gene expression this raises an interesting conundrum about the way fast cell proliferation and gene expression are coordinated.

The first step of gene expression involves a length of DNA being transcribed to produce an intermediate molecule called a messenger RNA (mRNA), which is then translated to produce a protein. However, some mRNA molecules contain regions called 'introns' that are not translated and must instead be removed via a time-consuming process called 'splicing' before the protein is produced.

At first a *Drosophila* embryo uses mRNA molecules that were spliced and packaged inside the egg by the mother, but later it starts to make its own mRNA molecules. The very first mRNA molecules made by the early embryo tend to be short and lack introns. The shortness of these molecules is thought to reflect the fact there is not enough time to produce longer mRNA molecules. Is the same 'need for speed' also responsible for the lack of introns in these molecules?

Now, Guilgur et al. have tested this hypothesis by manipulating a gene named *fandango*, which codes for part of the cellular machinery that removes introns from mRNA molecules, in fruit flies. These mutant fruit flies had less of the Fandango protein than wild-type flies and while they passed through the early stages of development normally, they later developed defects—such as abnormally shaped cells. Guilgur et al. revealed that *fandango* mutants fail to splice out the introns in the mRNA molecules that are made in the early embryo, whereas similar mRNA molecules from the mother were spliced as normal. Further experiments suggested that wild-type embryos struggled to correctly splice an untypical early gene that had multiple introns.

Together the findings of Guilgur et al. suggest that when nuclei (or cells) are dividing rapidly, there is a strong selective pressure to splice mRNA molecules quickly in the short time between the divisions. Furthermore, this pressure appears to have shaped the architecture of the earliest genes expressed in the *Drosophila* embryo, which is why the first mRNA molecules produced by the embryo itself tend not to contain introns.

In yeast, *Drosophila*, and human cells, pre-mRNA splicing is mostly co-transcriptional (*Ameur et al., 2011*; *Khodor et al., 2011*), with in vivo splicing rates being in the order of 30 s to approximately 3 min once the intron is transcribed (*Alexander et al., 2010*; *Huranova et al., 2010*; *Schmidt et al., 2011*). As most early zygotic transcripts are intronless (*De Renzis et al., 2007*), syncytial blastoderm interphases can be as short as 4 to 5 min, and given the fact that mitosis inhibits splicing (*Shin and Manley, 2002*), we hypothesized that further to the selective pressure for small transcriptional units, there is also a pressure to avoid pre-mRNA splicing during early zygotic expression. We isolated two mutant alleles for a subunit of the NTC/Prp19 complexes, known to be important for efficient spliceosome activation, which specifically impaired pre-mRNA splicing of early zygotic but not maternally encoded transcripts. We showed that the differential splicing defects were not related to any particular structure/sequence of the early zygotic transcripts or differential association of spliceosomal components to the NTC/Prp19 complexes. Ectopic maternal expression of an early zygotic transcript in a mutant background was sufficient to suppress its splicing defects, suggesting that they were dependent on the developmental context of gene expression. We reasoned that constraints on pre-mRNA splicing are present during *Drosophila* early embryonic development. Consistently, a small early zygotic transcript with four introns was poorly spliced in wild-type embryos. Such constraints on pre-mRNA splicing are a likely explanation why most early zygotic genes are intronless and suggest that highly proliferative tissues need coordination between cell cycle and gene architecture for correct gene expression and avoidance of abnormally processed transcripts. Our results strongly argue in

favor of a developmental pre-requisite for highly efficient splicing during fast development, therefore we propose that the requirement for overall splicing efficiency is likely to vary during development.

## Results and discussion

### *Drosophila* Fandango/Xab2 is required for blastoderm cellularization

Previously we isolated a collection of maternal mutants defective in blastoderm cellularization and/or germ-band extension (*Pimenta-Marques et al., 2008*). Complementation group 7 contained two different mutant alleles with similar defects in blastoderm cellularization. Through deficiency mapping and a candidate gene approach we concluded that both were allelic to the uncharacterized coding gene CG6197 (Flybase). To confirm the mutants' identity, we rescued their zygotic lethality, female sterility (germ-line clones), and blastoderm cellularization defects (maternal mutant embryos) using a genomic fragment construct that contained a wild-type copy of CG6197 (*Figure 1—figure supplement 1A*, data not shown). Both isolated alleles of CG6197 showed identical phenotypes: maternal mutant embryos (hereafter referred to as mutant embryos) showed normal syncytial nuclear divisions (*Figure 1A,B*) but subsequently failed to elongate the cortical nuclei, which became mislocalized during blastoderm cellularization (*Figure 1C–F*, quantification in *Figure 1G*). The blastoderm cellularization phenotype was remarkably similar to that described for *kugelkern/charleston* mutant embryos (*Brandt et al., 2006*; *Pilot et al., 2006*). Based on the observed phenotypes, we named the corresponding gene *fandango*, after the Iberian folk dance.

*fandango* encodes the *Drosophila* ortholog of yeast SYF1 (synthetic lethal with cdc41) (*Russell et al., 2000*) and human XAB2 (XPA binding protein 2) (*Nakatsu et al., 2000*; *Kuraoka et al., 2008*). These proteins were described as subunits of the NTC/Prp19 complexes, which are important for spliceosome stabilization and activation (*Chan et al., 2003*; *Chang et al., 2009*; *Hogg et al., 2010*). Fandango protein has multiple tetratricopeptide repeat (TPR) motifs, which is a protein–protein interaction module (*Zeytuni and Zarivach, 2012*). Sequencing both alleles of *fandango* (*fand¹* and *fand²*) revealed distinct mutations within the *fandango* open reading frame (ORF). *fand¹* contained a missense point mutation in a highly conserved residue within TPR domain VII (from an alanine to a valine; A401V), whereas *fand²* contained a microdeletion of 18 nucleotides within TPR domain VI, which deleted six conserved amino acids from position 355 to 360 (*Figure 1—figure supplement 1B*). In total protein extracts, both *fand¹* and *fand²* mutant embryos showed a significant reduction in Fandango protein levels compared to control (*Figure 1I*). fandango mRNA levels, analyzed by real-time qPCR, were similar between control and *fand¹* mutant embryos (*Figure 1J*), suggesting that the mutation did not alter the stability of the encoding pre-mRNA.

### *Drosophila* Fandango/Xab2 is differentially required for splicing of maternal and early zygotic pre-mRNAs

As noted above *fandango* maternal mutant embryos and *kugelkern* (*kuk*) mutant embryos showed remarkably similar blastoderm cellularization defects (*Brandt et al., 2006*; *Pilot et al., 2006*). Since *fandango* encodes a protein whose yeast and human orthologs are required for efficient spliceosome activity, we hypothesized that Fandango was required for splicing of *kuk* transcripts. *kuk* encodes two different transcripts, which vary in intron size (*Figure 2A*). Both transcripts are predicted to encode the same protein. Analysis of publicly available modENCODE transcriptome datasets (*Graveley et al., 2011*) suggested that the large *kuk* transcript was maternally expressed, whereas the small *kuk* transcript was only expressed zygotically. Through RT-PCR analysis we confirmed that similarly to control maternal genes (*nanos* and *oskar*) the large *kuk* transcript was maternally expressed (being present in unfertilized eggs), whereas the small *kuk* transcript was exclusively zygotically expressed (being present only in fertilized eggs) as the case of well-known early zygotic genes (*even-skipped* and *krüppel*) (*Figure 2—figure supplement 1A*).

To investigate by RT-PCR whether Fandango was required for splicing of *kuk* pre-mRNAs, specific sets of primers (exon–exon, e–e; intron–exon, i–e) were designed for each *kuk* transcript, taking advantage of a longer 3'UTR in the small *kuk* transcript (*Figure 2A*). Surprisingly, whereas *fandango* embryos showed significant splicing defects of the small zygotic *kuk* transcript, the large maternal *kuk* transcript was correctly spliced (*Figure 2B*; *Figure 2—figure supplement 1B*). Splicing defects were fully rescued by a genomic fragment construct that contained a wild type copy of *fandango* (*Figure 2—figure supplement 1C*). The differential requirement of Fandango for splicing of *kuk* transcripts prompted us

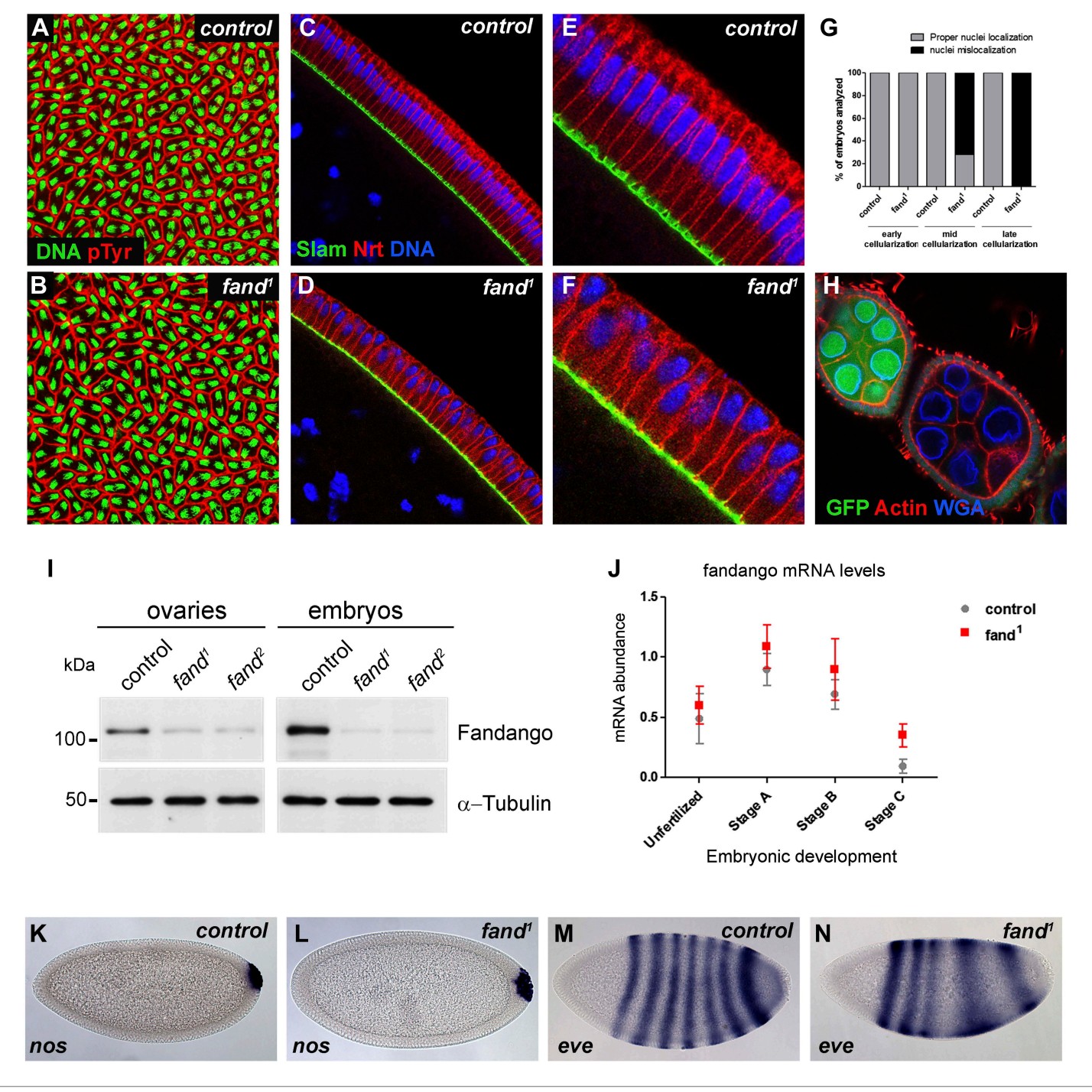

**Figure 1**. Drosophila Fandango/Xab2 is required for blastoderm cellularization. (**A** and **B**) Panels show embryos with normal syncytial blastoderm nuclear divisions in control embryos (*hs*-FLP; FRT42B) (**A**) and *fand¹* germ-line clone embryos (*hs*-FLP; FRT42B *fand¹*, maternal mutant) (**B**). Embryos were stained for DNA (green) and p-Tyrosine (red). (**C–F**) Panels show blastoderm cellularized embryos. Control embryos showed normal epithelial architecture with elongated nuclei and columnar cell shape (**C**). *fand¹* germ-line clone mutant embryos showed abnormal epithelial architecture, the cortical nuclei failed to elongate and became mislocalized (**D**). (**E** and **F**) Magnification of **C** and **D,** respectively. Embryos were stained for Slam (green), Neurotactin (red), and DNA (blue). (**G**) Quantification of *fandango* maternal mutant embryo phenotype during blastoderm cellularization. Early cellularization: control: 100% normal (n = 44), *fand¹*: 100% normal (n = 49); mid cellularization: control: 100% normal (n = 25), *fand¹*: 28% normal (n = 21); late cellularization: control: 100% normal (n = 42), *fand¹*: 0% normal (n = 38). (**H**) Maternally controlled oogenesis was normal in *fandango* mutant clones. Absence of endogenous nGFP (green) indicated that the cells were homozygous for *fand¹* mutation. Ovaries were stained for F-actin (red) and WGA (blue). (**I**) Western blot of
*Figure 1. Continued on next page*

*Figure 1. Continued*

whole protein extracts from embryos and ovaries mutant for *fand¹* and *fand²* alleles (germ-line clones) showed a clear reduction in Fandango protein levels compared to control tissues. It should be noticed that due to experimental constraints the total protein extracts from mutant ovaries included not only signal from mutant germ-line cells (homozygous for *fand¹*), but also the tightly associated heterozygote somatic follicle cells. α-Tubulin was used as a loading control. (**J**) Real-time qPCR analysis showed no significant differences in fandango mRNA levels between control and *fand¹* embryos during development (Two-way ANOVA p>0.05 ns.). fandango mRNA levels were normalized with β-actin mRNA levels. (**K–N**) in situ hybridization for *nanos* RNA (maternal) and *even-skipped* RNA (early zygotic) in blastoderm cellularized embryos. Both control (**K**) and *fand¹* mutant (**L**) embryos showed normal *nos* localization pattern in the pole cells. *fand¹* embryos (**N**) showed A–P patterning defects of *eve* compared to control embryos (**M**).

The following figure supplements are available for figure 1:

**Figure supplement 1**. *fandango* mutant alleles contain changes in highly conserved amino acids.

to investigate more than 20 other maternal and early zygotic genes. RT-PCR analysis of *fandango* embryos invariably showed splicing defects of early zygotic but not maternally encoded transcripts (*Figure 2C*, data not shown). High-throughput transcriptome sequencing (RNAseq) confirmed that splicing of early zygotic but not maternally encoded gene products was affected in *fandango* embryos (*Figure 2D*, *Figure 2—figure supplement 2A*). Maternal transcripts, whose intron size was equivalent to those observed in early zygotic transcripts, were unaffected (*Figure 2—figure supplement 2B*), which showed that Fandango was not specifically rate limiting for splicing of small introns. Comparison analysis of 5′ and 3′ splice site consensus sequences between maternal and zygotic pre-mRNA transcripts showed no significant differences (*Figure 2—figure supplement 2C*) and the two populations of transcripts displayed a similarly heterogeneous exon–intron structure (*Figure 2—figure supplement 2D*). RT-PCR analysis of maternally encoded transcripts from wild-type and *fandango* mutant ovaries (germ-line clones) also failed to detect splicing defects (*Figure 2—figure supplement 1D*). This suggested that the absence of splicing defects of maternally encoded transcripts in *fandango* embryos was not due to specific degradation of unspliced transcripts during oogenesis.

The differential requirement of Fandango for splicing of early zygotic encoded transcripts is fully consistent with the observation that maternally controlled oogenesis, primordial germ-cell formation, and syncytial nuclear divisions were normal in *fandango* mutants (*Figure 1A,B,H,K,L*), whereas the first detectable phenotype only occurred during zygotically controlled blastoderm cellularization (*Figure 1C–F*). Despite the fact that clonal analysis of the female germ line for both alleles of *fandango* showed normal oogenesis and egg laying (*Figure 1H*) (data not shown), Fandango protein levels were significantly reduced in the mutant ovaries (germ-line clones) (*Figure 1I*). *fandango* embryos also failed to initiate germ-band extension after blastoderm cellularization (data not shown). It was previously shown that anterior–posterior (A–P) patterning is required for germ-band extension (*Zallen and Wieschaus, 2004*). Consistently, *fandango* embryos showed A–P patterning defects in the early zygotic pair-rule gene *even-skipped* (*Figure 1M,N*).

## Fandango is similarly associated with the NTC/Prp19 complexes during oogenesis and early embryonic development

The highly conserved NTC/Prp19 and NTC/Prp19-related complexes are essential for pre-mRNA splicing as they facilitate the formation and progression between distinct spliceosome conformations during the splicing reaction (*Chan et al., 2003*; *Hogg et al., 2010*).

Endogenous Fandango and Prp19 physically interacted in the early embryo (*Figure 3A*). Moreover, both endogenous Fandango and Prp19 physically interacted with endogenous ISY1 and CDC5L (*Figure 3A*), confirming that Fandango is a *bona fide* subunit of *Drosophila* NTC/Prp19 complexes. Immunoprecipitation of Myc-tagged Fandango and Myc-tagged Prp19 from embryonic protein extracts also identified an identical group of interacting proteins (*Table 1*; *Supplementary file 1*). Whereas Myc-Fandango mostly interacted with the NTC/Prp19-related complex subunits, Myc-Prp19 interacted principally with the NTC/Prp19 complex subunits. This illustrated that, as in humans, distinct but interacting NTC/Prp19 complexes exist in *Drosophila*, in agreement with the recent suggestion that a remarkable degree of conservation of distinct splicing complexes exists among metazoans (*Herold et al., 2009*).

The differential requirements of Fandango for pre-mRNA splicing of maternal and early zygotic transcripts potentially suggest distinct interactions between Fandango and other splicing proteins

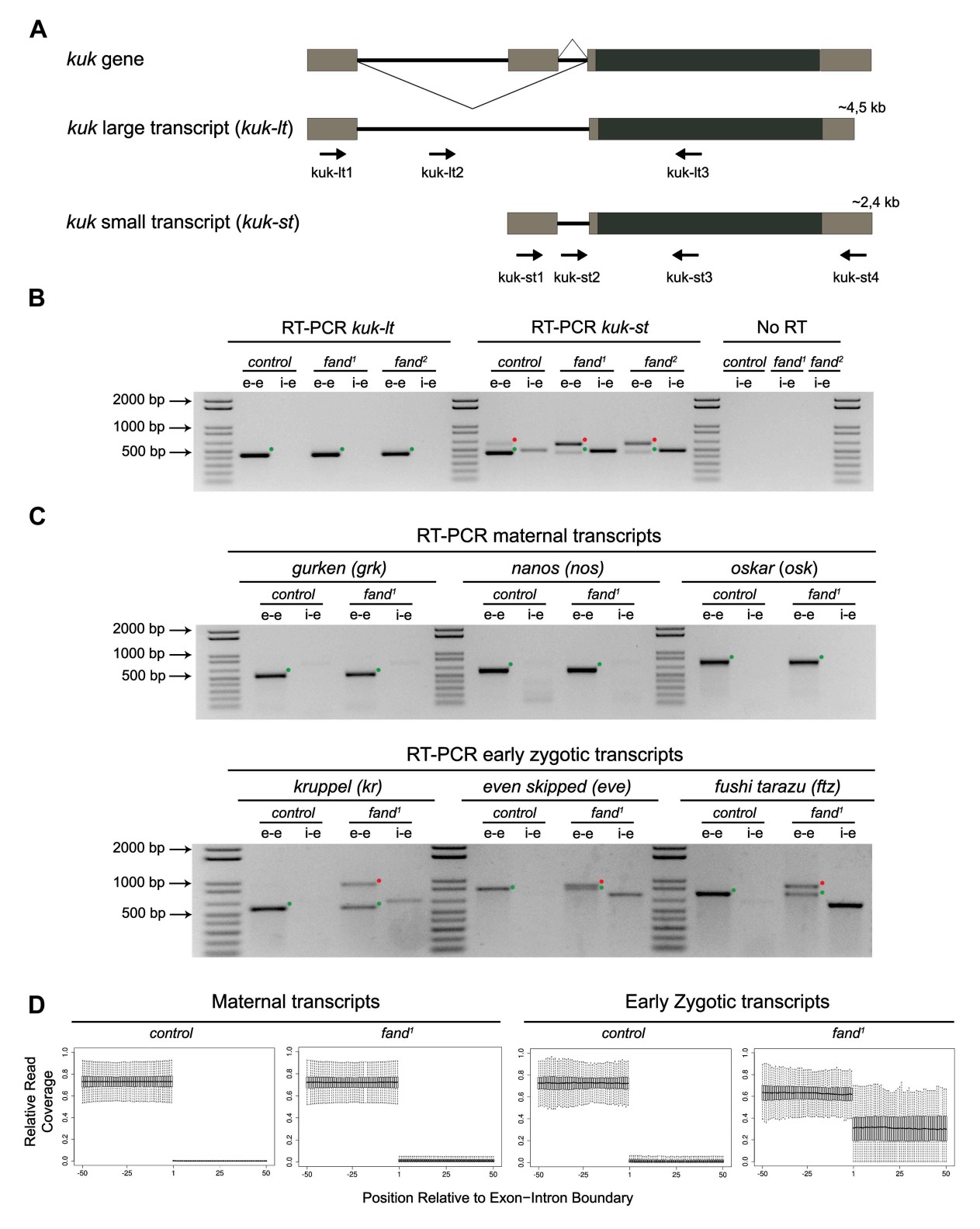

**Figure 2**. Splicing of early zygotic but not maternally encoded pre-mRNAs is affected in *fandango* mutants. (**A**) The *kugelkern* (*kuk*) locus encodes two transcripts of different size, *kuk-lt* containing a large intron and *kuk-st* with a short intron. Orientation and position of primers used for splicing analysis is indicated (arrows). (**B**) RT-PCR analysis of *kuk* transcripts. Control embryos yielded PCR products in the size predicted for the properly spliced forms of both *kuk* transcripts using exon–exon (e–e) primers (green dots, *kuk-lt*: 431 bp and *kuk-st*: 437 bp). *fandango* maternal mutant embryos
*Figure 2. Continued on next page*

*Figure 2. Continued*

(*fand¹* and *fand²* alleles) showed splicing defects only in the *kuk-st* transcript; PCR products were detected by e–e primers in the size expected for intron retention (red dots, *kuk-st*: 596 bp) and by intron–exon (i–e) primers (*kuk-st*: 474 bp). Splicing of the *kuk-lt* was not affected in *fandango* mutant background; PCR products were only detected with e–e primers in the predicted size for the correctly spliced pre-mRNA (green dots, *kuk-lt*: 431 bp). 'No RT' controls (only total RNA as template) yielded no amplification, meaning there was no contamination with genomic DNA in the samples tested. (**C**) RT-PCR analysis of maternal and early zygotic genes. Maternal transcripts were properly spliced, in both, control and *fand¹* mutant embryos; PCR products were only detected using e–e primers (green dots, *grk*: 527, *nos*: 581, *osk*: 762 bp). In contrast, early zygotic transcripts were correctly spliced only in control embryos (green dots, *kr*: 559, *eve*: 828, *ftz*: 753 bp). *fand¹* mutant embryos yielded PCR products in the size predicted for intron retention with e–e primers (red dots, *kr*: 932, *eve*: 899, *ftz*: 900 bp) and with i–e primers (*kr*: 629, *eve*: 720, *ftz*: 595 bp). All PCR bands showed in the panels were cloned and sequenced to confirm their identity. Green dots indicate correctly spliced transcripts, red dots indicate unspliced transcripts (intron retention). (**D**) RNA-Seq data confirmed that zygotic but not maternally encoded transcripts displayed a large fraction of splicing defects (intron retention) in *fand¹* mutant embryos. The panel shows box plot of the distribution of numbers of reads per bp relative to the total number of reads falling inside a 100 bp window centered around the 5'splice sites of zygotic (n = 408 splice sites from 270 genes) or maternal genes (n = 5876 splice sites from 2048 genes).

The following figure supplements are available for figure 2:

**Figure supplement 1**. Splicing of early zygotic but not maternally encoded pre-mRNAs is affected in *fandango* mutants.

**Figure supplement 2**. Early zygotic but not maternally encoded pre-mRNAs shows significant intron retention in *fandango* mutants.

during oogenesis and early embryonic development. Nevertheless, immunoprecipitation of Myc-Fandango specifically expressed in the female germ line during oogenesis and in the early embryo identified a virtually identical group of interacting proteins: mostly subunits of the NTC/Prp19-related complex, and to a lesser extent, subunits of the NTC/Prp19 complex (*Table 1*; *Supplementary file 1*). These results showed that Fandango physically interacts with a similar group of splicing proteins during oogenesis and in the early embryo.

To better understand the splicing defects observed in *fandango* embryos, we investigated if the integrity of NTC/Prp19 complexes was affected in this mutant. Size-exclusion chromatography showed detectable changes in the integrity of NTC/Prp19 complexes in *fandango* embryos (*Figure 3B*), with a significant reduction in the levels of ISY1 protein (*Figure 3C*). ISY1 is a NTC/Prp19-related complex subunit (*Figure 3A*). The loss of integrity of the ISY1-positive ~600–800 kDa NTC/Prp19 complex (*Figure 3B*) and concomitant reduction in the stability of some of their subunits, most likely impaired efficient activation of the spliceosome (*Villa and Guthrie, 2005*) and were likely explanations for the splicing defects observed in *fandango* embryos. In agreement with the suboptimal spliceosome activation hypothesis, intron retention was the main splicing defect of early zygotic transcripts in *fandango* embryos (*Figure 2B,C*, *Figure 2—figure supplement 2B*; data not shown).

Levels of ISY1 were similarly affected in *fandango* mutants during oogenesis and in the early embryo (*Figure 3C*), suggesting this decrease did not explain the differential requirements of Fandango for splicing of early zygotic and maternally encoded transcripts. Mutant clonal analysis of a stronger allele of *fandango* (nonsense mutation), showed a complete loss of the female germ line in adult ovaries (data not shown). This demonstrated that the two isolated alleles of *fandango* are hypomorphic and suggested that Fandango was required, albeit at lower levels, for splicing of maternal transcripts. We concluded it is unlikely that a differential expression and/or association of core components of the spliceosome could potentially explain the differential requirements for Fandango between oogenesis and the early embryo. The most likely explanation is that Fandango is quantitatively (but not qualitatively) differentially required during early embryonic development.

## Reduction in Fandango levels affects mainly its splicing function

Transcriptional elongation can affect co-transcriptional splicing (*de la Mata et al., 2003*; *Ip et al., 2011*). It was recently shown that Syf1, the yeast ortholog of Fandango, is also important for RNApol II transcriptional activity (*Chanarat et al., 2011*; *David et al., 2011*), therefore we decided to investigate transcription in *fandango* embryos. Three intronless early zygotic genes (*nullo*, *snail*, and *scute*) and two early zygotic genes with introns (*even-skipped* and *tailless*) were selected for further analysis by real-time qPCR. During mid/late-syncytial blastoderm (stage B) (*Figure 4A*, 'Materials and methods'), no significant differences in transcript abundance were observed between control and

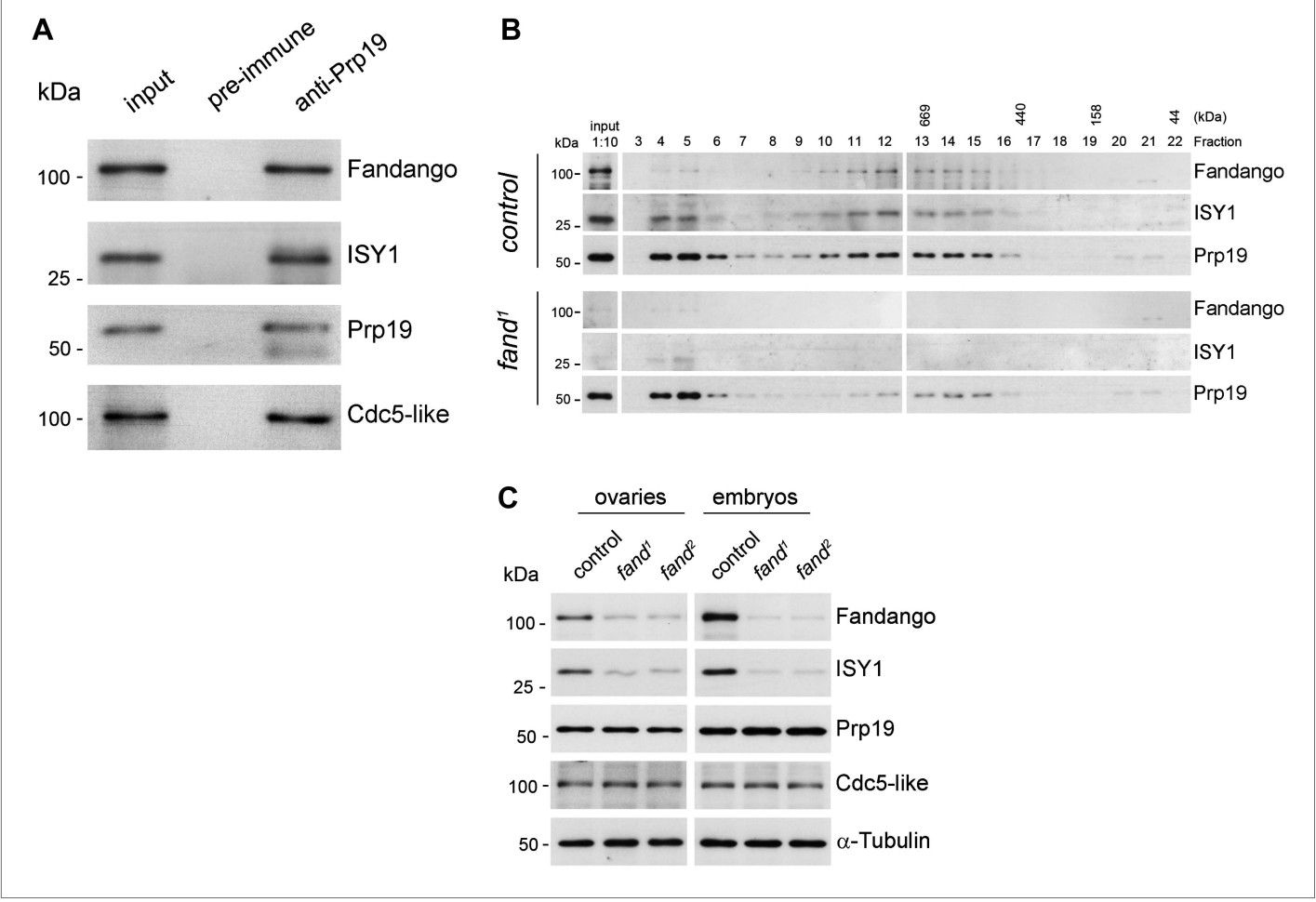

**Figure 3**. Fandango physically interacts with a similar group of splicing proteins during oogenesis and embryogenesis. (**A**) Pull down assay from nuclear-enriched protein extracts using a polyclonal antibody of Prp19. Endogenous Prp19 interacts physically with Fandango and other subunits of the NTC/Prp19 complexes (ISY1 and CDC5L). Pre-immune serum was used in the control. (**B**) Size-exclusion chromatography of control and *fand¹* mutant protein extracts from 0–3 hr embryo collections using a Superose 6 10/300 column. After separation, each fraction was analyzed by Western blot. NTC/Prp19 complexes subunits (Prp19, Fandango, and ISY1) were part of a ~600–800 kDa complex and also co-purified in a significantly larger complex (fraction 4 and 5). *fand¹* mutant protein extracts showed a significant reduction in levels of Fandango and ISY1 subunits and a size reduction of the Prp19-positive ~600–800 kDa complex. (**C**) Western-blot analysis of total protein extracts from ovaries (left) and 0–3 hr embryos (right) from control and both *fandango* alleles, showed a reduction of Fandango and ISY1 protein levels in both tissues. Protein levels of Prp19 and CDC5L were not affected. α-Tubulin was used as loading control. Fandango Western blot is the same as shown in *Figure 1I*.

fandango (*Figure 4—figure supplement 1A*), whereas embryos mutant for *grapes* showed the expected reduction of transcript levels (*Figure 4—figure supplement 1A*; *Sibon et al., 1997*).

During transcriptional elongation, RNApol II is specifically phosphorylated on the Ser2 residue of its carboxy-terminal domain (CTD) (*Hsin and Manley, 2012*). In agreement with the onset of early zygotic transcription, we observed a significant increase in RNApol II CTD Ser2 phosphorylation as the embryo developed from early/mid-syncytial blastoderm (stage A), into mid/late-syncytial blastoderm (stage B), and blastoderm cellularization (interphase 14) (stage C) (*Figure 4A,B*). Both control and *fandango* embryos showed a similar increase in global levels of RNApol II CTD Ser2 phosphorylation (*Figure 4B,C*). As transcriptional regulation during interphase 14 (stage C) relies on correct expression of early zygotic genes and degradation of many maternal RNAs (MZT) (*Tadros and Lipshitz, 2009*), we concluded that transcriptional changes at this stage (*Figure 4—figure supplement 1A*) were most likely a consequence of the widespread defects occurring during mid/late-syncytial blastoderm. Altogether, we concluded that the observed reduction in Fandango levels affects mainly its splicing function.

**Table 1.** LC-MS analysis of co-immunoprecipitation assays from ovaries and embryos

| Drosophila | | | Fandango-myc | | | | Prp19-myc | |
| --- | --- | --- | --- | --- | --- | --- | --- | --- |
| CG | gene | | ovaries | | embryos | | embryos | |
| | | Human/yeast | rep1 | rep2 | rep1 | rep2 | rep1 | rep2 |
| prp19 complex | | | | | | | | |
| CG5519 | prp19 | PRP19/Prp19 | + | + | + | + | +++ | ++ |
| CG6905 | cdc5-like | CDC5L/Cef1 | + | + | + | + | +++ | ++ |
| CG1796 | Tango4 | PLRG1/Prp46 | + | + | + | + | + | + |
| CG4980 | - | BCAS2/Snt309 | – | + | - | - | + | + |
| CG12135 | c12.1 | CWC15/cwc15 | + | – | – | - | – | – |
| Prp19 related | | | | | | | | |
| CG6197 | Fandango | Xab2/Syf1 | +++ | +++ | +++ | +++ | + | + |
| CG31368 | – | AQR/– | +++ | +++ | +++ | +++ | + | + |
| CG4886 | cyp33 | PPIE/– | ++ | ++ | + | ++ | + | + |
| CG9667 | – | ISY1/ISY1 | + | + | + | + | + | + |
| CG8264 | Bx42 | SNW1/Prp45 | + | + | + | + | + | + |
| CG14641 | – | RBM22/Cwc2 | – | + | + | + | + | + |
| CG3193 | Crn | CRNKL1/Clf1 | – | + | – | – | + | – |
| CG13892 | cypl | PPIL1/- | – | – | – | + | – | – |
| CG1639 | l(1)10Bb | BUD31/Bud31 | – | – | – | – | + | + |

Co-immunoprecipitations were performed using total protein extracts from the different tissues expressing Myc-tagged Fandango or Myc-tagged-Prp19. Human and yeast homologues and the different sub-complexes are shown as described in (***Herold et al., 2009***). (–), (+), (++), (+++) correspond to 0, 1–9, 10–19, and >20 non-repeated peptides respectively. None of the proteins shown were detected in the negative controls (for detailed LC-MS analysis see ***Supplementary file 1***).

## Ectopic maternal expression of an early zygotic transcript in the mutant background was sufficient to suppress its splicing defects

To investigate if the differential requirement of Fandango for splicing of early zygotic and maternally encoded transcripts potentially resulted from distinct transcript sequences, we generated an early zygotic *kuk* transcript (*kuk-LacZ*) under the control of an UAS/Gal4 inducible promoter, where the open reading frame (ORF) was replaced by LacZ (***Figure 5A***, see 'Materials and methods' for more details). As expected, when this construct was expressed zygotically, it was correctly spliced in control but not in *fandango* embryos (***Figure 5B***). In contrast, splicing of the *kuk-LacZ* construct occurred normally in both control and *fandango* mutants when it was expressed maternally (***Figure 5B***). Since maternal expression of an early zygotic transcript, in a *fandango* mutant background, was enough to suppress its splicing defects, we concluded that the differential requirement of Fandango for splicing of early zygotic transcripts was most likely due to the developmental context of gene expression and not a particularity in the early zygotic pre-mRNA sequences. Consistently, we failed to detect differences related to intron size, splice sites consensus, and exon–intron structure between maternal and zygotic transcripts (***Figure 2—figure supplement 2B–D***).

## A small early zygotic transcript with multiple introns was poorly spliced in wild-type embryos

*fandango* mutants showed a significant reduction in Fandango and ISY1 protein levels (***Figure 3C***), which most likely impaired efficient activation of the spliceosome (***Villa and Guthrie, 2005***). Since mitosis inhibits splicing (***Shin and Manley, 2002***), pre-mRNA splicing of early zygotic transcripts needs to be highly efficient for these genes to be correctly expressed. This suggests the existence of a developmental pre-requisite for highly efficient splicing, so that a suboptimal activation of the spliceosome would specifically impair pre-mRNA splicing of early zygotic but not maternal transcripts. Wild-type embryos already showed a detectable amount of intron retention in early zygotic transcripts

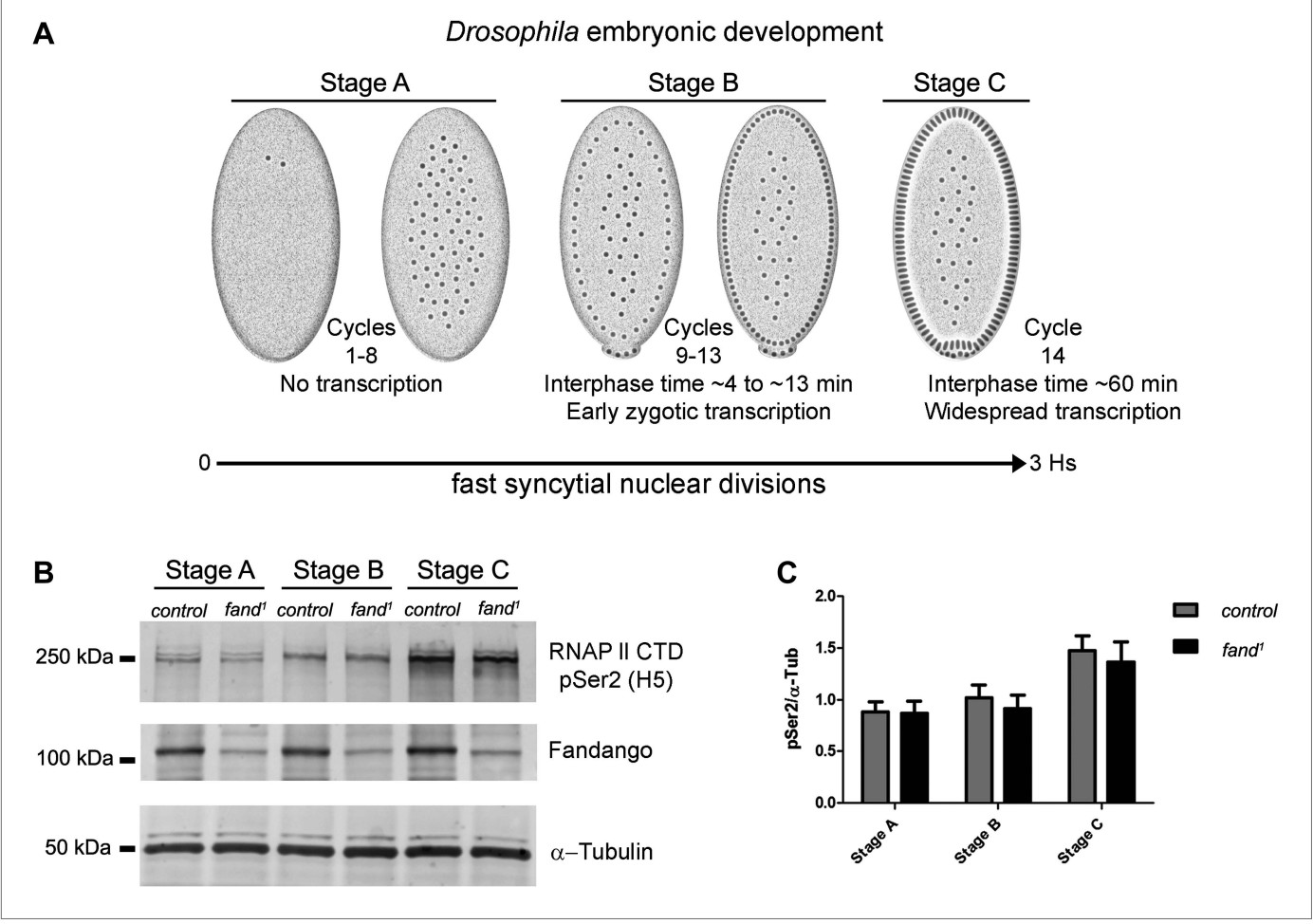

**Figure 4**. Early zygotic transcription is not affected during mid/late-syncytial blastoderm in *fandango* mutants. (**A**) Embryos were divided into three different groups according to developmental stage ('Materials and methods'), stage A: early/mid-syncytial blastoderm embryos, stage B: mid/late-syncytial blastoderm embryos, and stage C: blastoderm cellularization embryos. (**B**) Western blot for RNApol II CTD Ser2 phosphorylation levels. Control and *fand¹* embryos showed a similar increase in the global levels of RNApol II CTD Ser2 phosphorylation over the course of early embryonic development. α-Tubulin was used as a loading control. (**C**) Quantification of the CTD Ser2 phosphorylation from five independent western blot assays showed no significant difference at any of the embryonic developmental stages analyzed (Two-way ANOVA $p > 0.05$ ns.).

The following figure supplements are available for figure 4:

**Figure supplement 1**. Early zygotic transcription is not affected during mid/late-syncytial blastoderm in *fandango* mutants.

(*Figure 2B,D*, *Figure 2—figure supplement 2B*), which was dramatically exacerbated in *fandango* embryos (*Figure 2B,D*, *Figure 2—figure supplement 2B*).

We hypothesized that regardless of transcript size, there was also a constraint on pre-mRNA splicing of early zygotic transcripts in wild-type embryos. We generated a gene where the 5'UTR sequence including the intron of the small zygotic *kuk* transcript was quadruplicate to test this hypothesis (*Figure 6A,D* see 'Materials and methods' for more details). Quadruplicate introns were linked by in-frame LacZ coding sequences, and the entire construct (*4x intron kuk-LacZ*) was under the control either of an endogenous early zygotic minimal promoter (*nullo-4x intron kuk-LacZ*, ~2.5 Kb) (*Figure 6A*) or an inducible UAS/Gal4 promoter (*UAS-4x intron kuk-LacZ*, ~2.5 Kb) (*Figure 6D*). The total size of the encoded pre-mRNAs was comparable to many other endogenous early zygotic genes (e.g., *kugelkern*, *runt*, *krupple*). As a control, we introduced point mutations in the splice sites of these constructs to generate comparable intronless transcripts (*no intron kuk-LacZ*) (*Figure 6A,D*).

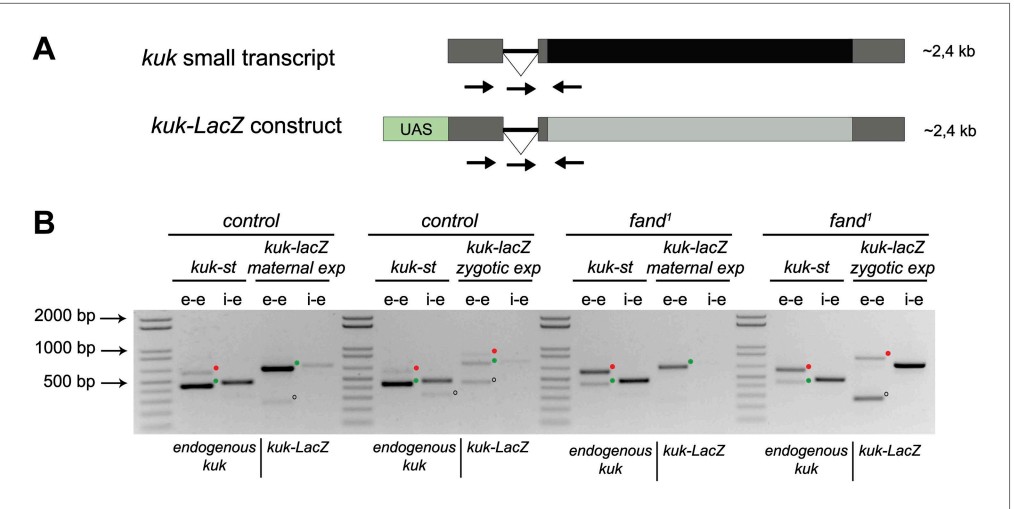

**Figure 5**. Ectopic maternal expression of an early zygotic transcript in the mutant background is sufficient to suppress its splicing defects. (**A**) The *kuk-LacZ* construct was built using the 5'UTR, the intron and the 3'UTR of the *kuk* small transcript (dark gray), and replacing the *kuk* ORF (black) by the LacZ coding sequence (light gray). To induce the expression of this construct it was put under the control of the UAS promoter (green) to drive the tissue specific expression with GAL4 drivers. Orientation and position of primers used for splicing analysis is indicated (arrows). (**B**) RT-PCR analysis of the *kuk-LacZ* construct. When it was zygotically expressed, it was correctly spliced in control but not in *fand¹* embryos (similarly to the endogenous small *kuk* transcript). Intron retention with e–e primers (red dots, *kuk-st*: 596 bp and *kuk-LacZ*: 869 bp) and a PCR product with i–e primers (751 bp) were observed in the mutant. When it was maternally expressed, *kuk-LacZ* construct was correctly spliced both in control and *fand¹* embryos, being detected just the spliced form of the construct (green dots, *kuk-st*: 437 bp and *kuk-LacZ*: 713 bp). In contrast, the endogenous zygotically expressed small *kuk* transcript (*kuk-st*) is still poorly spliced in *fand¹* embryos carrying the *kuk-LacZ* construct. Open circles indicate unspecific PCR products (confirmed by sequencing). Green dots indicate correctly spliced transcripts, whereas red dots indicate unspliced transcripts (intron retention).

Only the first intron (int1) of the *4x intron kuk-LacZ* construct was correctly spliced when it was zygotically expressed in wild-type embryos under the control of an endogenous early zygotic minimal promoter (*Figure 6B*). Likewise, when the *4x intron kuk-LacZ* construct was early zygotically expressed under the control of the inducible promoter UAS/Gal4 there were similar splicing defects (intron retention) (*Figure 6E*).

Measurement of in vivo kinetics of mRNA splicing showed that half-lives for splicing reactions are <1 min for the first intron, but 2–8 min for both second and third introns (*Audibert et al., 2002*). Hence, splicing of two or more introns requires more time than transcription and becomes rate limiting. Consistent with the hypothesis of a temporal constraint on pre-mRNA splicing, when the *4x intron kuk-LacZ* construct was zygotically expressed, the splicing defects of the firstly transcribed 5'-localized introns (Int1 and Int2) were significantly weaker than those observed in the later transcribed 3'-localized introns (Int3 and Int4) (*Figure 6E*). Importantly, maternal expression of this construct was sufficient to significantly suppress its splicing defects (*Figure 6E*). Real-time qPCR analysis showed that these constructs were equivalently zygotically and maternally expressed (*Figure 6C,F*). This suggested that splicing did not quantitatively impair early zygotic transcription, which was consistent with the observation that the rates of transcriptional elongation proceed independently of splicing (*Brody et al., 2011*).

We showed that in wild-type embryos pre-mRNA splicing imposed significant constraints on early zygotic expression, which is a likely explanation why most early zygotic genes are intronless (*De Renzis et al., 2007*). Although a moderate decrease in the length of syncytial blastoderm interphases (seen in *grapes* mutant embryos [*Sibon et al., 1997*]) was not sufficient to induce splicing defects in otherwise wild-type embryos (data not shown), we hypothesize that avoidance of pre-mRNA splicing during early zygotic expression is a consequence of the extremely short interphases and frequent mitotic cycles. Similarly to *Drosophila*, mosquito *Aedes aegypti* and the zebrafish *Danio rerio* early zygotic transcripts

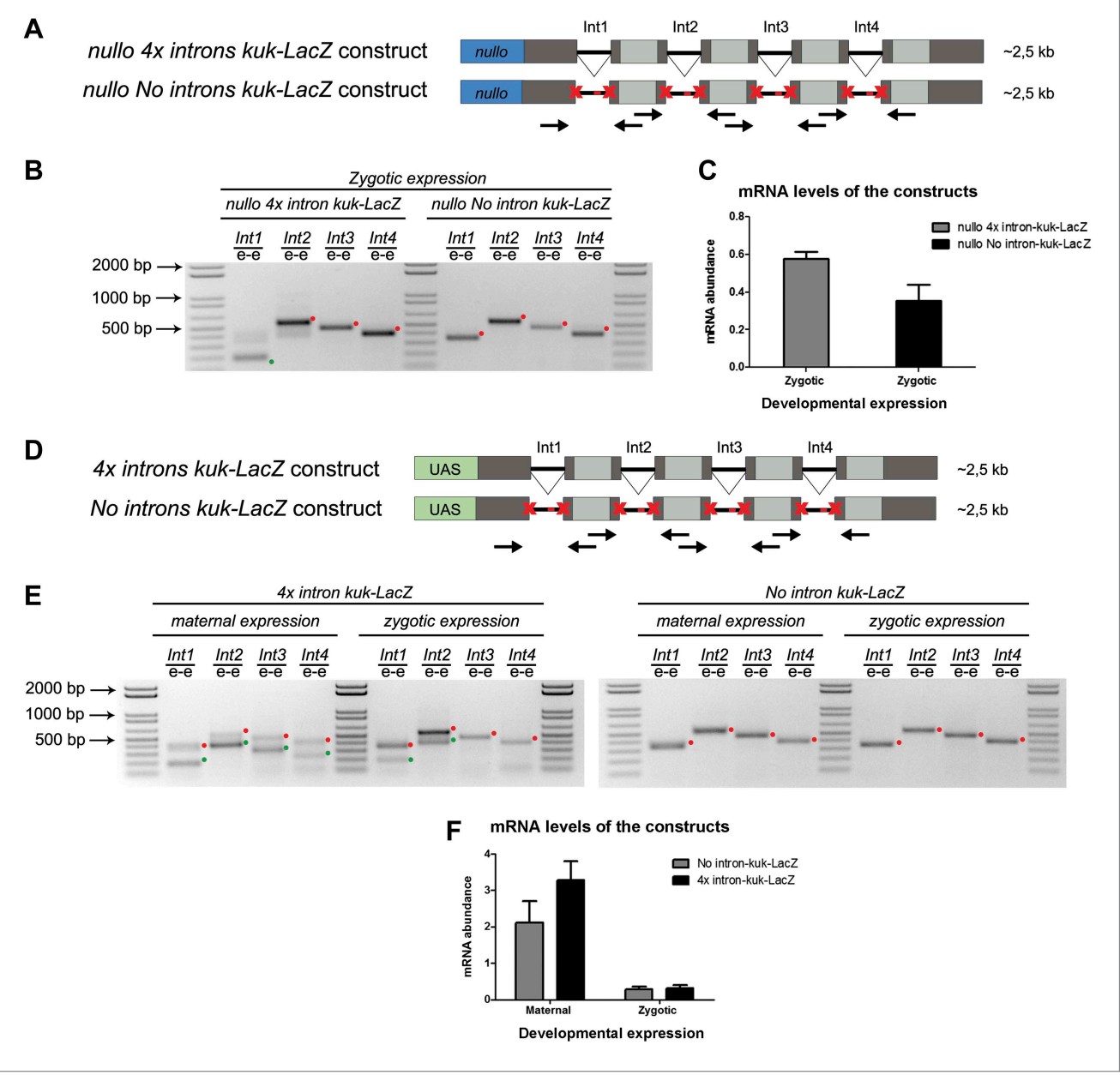

**Figure 6**. A small early zygotic transcript containing four introns is poorly spliced in wild-type embryos. (**A** and **D**) The *4x intron kuk-LacZ* construct was a variant of the *kuk-LacZ* that contains four copies of *kuk* small transcript intron (dark gray). Each intron is separated by 201 nucleotides of an in frame Lac-Z sequence (light gray). The *no intron kuk-LacZ* construct has all splice sites present in the *4x intron kuk-LacZ* construct mutated to thymidines. The constructs were fused to a *nullo* minimal promoter (blue) (**A**), or fused to an inducible UAS promoter (green) (**D**). Orientation and position of primers used for splicing analysis is indicated (arrows). (**B**) RT-PCR analysis showed significant splicing defects (intron retention) of the *4x intron kuk-LacZ* construct when expressed under the control of an endogenous early zygotic promoter (*nullo* promoter). The first intron was correctly spliced, being detected mainly the PCR product corresponding to the spliced form (green dot). The remaining introns (second, third, and fourth) were completely unspliced (red dots, intron retention). In the intronless (*no intron kuk-LacZ*) construct, under the control of the same *nullo* promoter were only observed PCR bands whose sizes correspond to unspliced forms (red dots, intron retention). (**C**) Real-time qPCR analysis showed that the *4x intron kuk-LacZ* and *no intron kuk-LacZ* constructs were expressed to the same extent when under the control of the *nullo* minimal promoter (*t* test p>0.05 ns.). (**E**) RT-PCR analysis of the *4x intron kuk-LacZ* construct showed significant splicing defects (intron retention) when zygotically expressed in wild-type embryos under the control of an inducible UAS promoter. Although the most 5'-localized introns (first and second) were still partially spliced, being observed two PCR bands corresponding to the spliced (green dots, int1: 191 and int2: 385 bp), and unspliced forms (red dots, int1: 347 and int2: 541 bp). The furthest 3'-localized introns (third and fourth) were completely unspliced, being only observed one PCR band with the size corresponding to intron retention (red dots, int3: 463 and int4: 385 bp). Maternal expression of the *4x intron kuk-LacZ* construct was sufficient to significantly suppress splicing defects in

*Figure 6. Continued on next page*

*Figure 6. Continued*

the four introns analyzed (green dots, spliced forms: int1: 191, int2: 385, int3: 307, int4: 229 bp; red dots, unspliced forms: int1: 347, int2: 541, int3: 463, int4: 385 bp). Zygotic and maternal expression of the *no intron kuk-LacZ* construct only showed PCR bands with sizes corresponding to unspliced forms (red dots, intron retention). (**F**) Real-time qPCR analysis showed that the *4x intron kuk-LacZ* and *no intron kuk-LacZ* constructs were expressed to the same extent both maternally (Two-way ANOVA p>0.05 ns.) and zygotically (p>0.05 ns.) in wild-type embryos. All PCR bands shown in these panels were cloned and sequenced to confirm their identity. Green dots indicate correctly spliced transcripts, red dots indicate unspliced transcripts (intron retention).

are frequently intronless when compared with the rest of the transcriptome (*Biedler et al., 2012*; *Heyn et al., 2014*). This suggests that highly proliferative tissues need coordination between cell cycle and gene architecture for correct gene expression and avoidance of abnormally processed transcripts.

Our results highlight cell cycle constraints during early embryonic development as a force capable of driving changes in gene architecture of multicellular organisms. In unicellular organisms intron paucity correlates with a bias toward the 5′ ends, whereas introns from multicellular genomes are evenly distributed throughout the genes (*Mourier and Jeffares, 2003*). This suggests that similar constraints on gene architecture are also likely to exist in yeast and other fast-dividing single cell eukaryotes.

The way splicing efficiency might be regulated through changes in constitutive spliceosome factors and how it might influence differential gene expression is a new area of interest. In this study, we present experimental evidence supporting the hypothesis of a requirement for highly efficient pre-mRNA splicing during early embryonic development. Since the NTC/Prp19 complexes are known to be important for efficient spliceosome activation, and our mutant alleles specifically impaired pre-mRNA splicing of early zygotic but not maternally encoded transcripts, we propose that overall requirements for splicing efficiency are likely to vary during development, being the NTC/Prp19 complexes a key modulator of spliceosome activation rates. In agreement with this hypothesis, Prp19 expression varies during neuronal differentiation (*Urano et al., 2006*).

In plants it was recently shown that removal of retained introns regulates translation in rapidly developing gametophytes (*Boothby et al., 2013*). In *Drosophila*, a sub-population of early zygotic transcripts with introns similarly showed some degree of intron retention in wild-type embryos (*Figure 2B,D*, *Figure 2—figure supplement 2B*). Our results also suggest that the pre-requisite for highly efficient splicing during early embryonic development is paradoxically also likely to play an important regulatory role in the expression of a subset of early zygotic transcripts, which further supports the possibility that modulation of spliceosome activation per se is important for differential regulation of gene expression during development.

## Materials and methods

### Fly work and genetics

Flies were raised using standard techniques. The *fandango* alleles were isolated in a previously reported maternal screen (*Pimenta-Marques et al., 2008*). Maternal mutant embryos and germ-line mutant clones were generated using the FLP/FRT *ovo^D* system (*Chou and Perrimon, 1992*). Germ-line clones of *fand^1* and *fand^2* were established by crossing FRT42B *fand^1*/CyO or FRT42B *fand^2*/CyO virgins to *hs*-FLP; FRT42B *ovo^D*/CyO males and the progeny was heat shocked once at 37°C for 1 hr during second and third larval instar stages. As control we generated germ-line clones with FRT42B by crossing FRT42B/CyO virgins to *hs*-FLP; FRT42B *ovo^D*/CyO males and followed by the heat shock procedure described before.

To generate homozygous mutant clones in ovaries for *fand^1* (negative for nuclear GFP label, nGFPminus) we used FLP/FRT to induce mitotic recombination. Females *y, w, hs*-FLP; FRT42B nGFP/CyO *hs-hid* flies were crossed with *w*; FRT42B, *fand^1*/CyO *hs-hid* males. Recombination was induced by 1-hr heat shock at 37°C at second and third instar larval stage. Adult ovaries were harvested from 4–5-day-old females and subsequently processed for immunofluorescence.

Viability and phenotypes of *fandango* alleles were complemented by crossing a transgenic fly carrying a genomic fragment construct that contained a wild-type copy of CG6197 (*wt-fandango*). w; FRT42B, *fand^1*/CyO virgins were crossed to *wt-fandango*; FRT42B, *fand^2*/CyO males; reciprocal crosses were also performed. Offspring were counted to determine viability. Rescue of maternal phenotypes

(cellularized blastoderm defects and splicing defects in early zygotic transcripts) was also analyzed in embryos laid by F1 *wt-fandango/ +; FRT42B, fand¹/FRT42B fand²* females. Germ-line clones of *fand¹* and *fand²* were also rescued (cellularized blastoderm defects and splicing defects in early zygotic transcripts) by a copy of *wt-fandango* in the third chromosome. FRT42B *fand¹/CyO; wt-fandango* or FRT42B *fand²/CyO; wt-fandango* virgins were crossed with *hs*-FLP; FRT42B *ovoᴰ/CyO* males and heat shock performed as described above.

To induce maternal and zygotic expression of the *UAS-kuk-LacZ* construct in control and *fandango* maternal mutant embryos, we performed the following crosses:

Maternal expression in control genetic background: virgin females +/+; *Nanos-Gal4, UAS-kuk-LacZ/TM6B* crossed with wild-type males.

Zygotic expression in control genetic background: virgin females +/+; *actin-Gal4/TM6B* crossed with +/+; *UAS-kuk-LacZ* males.

Maternal expression in *fandango* maternal mutant genetic background: firstly, virgin females FRT42B *fand¹/CyO; Nanos-Gal4, UAS-kuk-LacZ/TM6B* crossed with *hs*-FLP; FRT42B *ovoᴰ/CyO* males, and heat shocked as described above. After eclosion, Cy⁺ and Tb⁺ females were selected from the progeny and crossed to wild-type males.

Zygotic expression in *fandango* maternal mutant genetic background: firstly, virgin females FRT42B *fand¹/CyO; actin-Gal4/TM6B* were crossed with *hs*-FLP; FRT42B *ovoᴰ/CyO* males, and heat shocked as described. After eclosion, Cy⁺ and Tb⁺ virgin females were selected from the progeny and crossed to +/+; *UAS-kuk-LacZ* males.

To induce maternal and zygotic expression of the *4x intron kuk-LacZ* and *no intron kuk-LacZ* constructs we performed following crosses:

Maternal expression: firstly, virgin females +/+; *actin-Gal4/TM6B* crossed with +/+; *UAS-4xintron-kuk-LacZ/TM6B* or *UAS-nointron-kuk-LacZ/TM6B* males. After eclosion, females Tb⁺ (+/+; *actin-Gal4/ UAS-4xintron-kuk-LacZ* or +/+; *actin-Gal4/UAS-nointron-kuk-LacZ*) were selected and crossed with wild-type males.

Zygotic expression: virgin females +/+; *actin-Gal4/TM6B* were crossed with +/+; *UAS-4xintron-kuk-LacZ/TM6B* or *UAS-nointron-kuk-LacZ/TM6B* males.

To analyze zygotic expression of the *4x intron kuk-LacZ* and *no intron kuk-LacZ* constructs under the control of the minimal promoter of the gene *nullo*, females carrying the corresponding construct were selected and crossed with wild-type males.

To drive embryonic and ovarian expression of Myc-tagged Fandango and Myc-tagged Prp19 proteins, *Nanos-Gal4* homozygous virgins were crossed with UAS-*Fandango-6xMyc* males or *UAS-Prp19-6xMyc/TM6B* males, respectively. After eclosion females (in case of Myc-Fandango) or Tb⁺ females (in case of Myc-Prp19) were selected, dissected ovaries from 4–5-day-old females, or laid embryos after a cross with wild-type males were used for protein extraction.

## Cloning of *fandango* alleles

To identify the gene responsible for lethality in *fandango* alleles, we performed a complementation analysis using the Bloomington 2R Deficiency kit.

Deficiency Df(2R)CX1 (covering an interval from cytological band 49C1 to 50D2, Bloomington stock number 442) failed to complement zygotic viability of both *fandango* alleles (complementation group 7). All additional 22 overlapping deficiencies complemented both *fandango* alleles. The cytological interval between bands 50B4-B6 (comprising 6 genes) was not covered by the 22 deficiencies. We cloned and sequenced these 6 genes from genomic DNA of both control and *fandango* alleles and identified mutations in gene CG6197 in both *fandango* alleles.

To confirm the identity of our mutants, we digested DNA from genomic clone (BACR14P04, Flybase) with restriction enzymes *Xbal* and *EcoRI* to generate a genomic fragment comprising the wild-type gene sequence of CG6197 (*wt-fandango*). Then we cloned the fragment into pCasper vector and used it to generate transgenic stocks (Bestgene, Chino Hills, CA, USA). A genomic wild-type copy of CG6197 under the control of its endogenous promoter fully complemented all known phenotypes in both *fandango* alleles.

## Immunohistochemistry

0–3 hr (after egg laying) embryos, both maternally mutant for *fandango* and control, were fixed and stained using standard procedures (*Pimenta-Marques et al., 2008*). For Neurotactin and Slam

immunostaining, the fixation procedure was modified: embryos were added to boiling heat fix solution (68 mM NaCl +0.1% Triton) and stirred for 1 min, then cooled by adding an equal volume of cooled fix solution. Immunostaining for oogenesis phenotypic analysis was performed as described in *Guilgur et al. (2012)*. Following primary antibodies used were: mouse anti-Neurotactin clone BP106 at 1:133 (DSHB, Iowa City, Iowa, USA); mouse anti-pTyr at 1:1000 (9411; Cell Signaling, Danvers, MA, USA), and rabbit anti-slam at 1:1000 (Ruth Lehman Lab). For F-actin staining, a 5-min incubation with phalloidin-Rhodamine at 1:200 dilution (Sigma, St Louis, MO, USA; stock concentration 1 mg/ml) was employed at room temperature. For DNA staining, we used SYTOX Green (Invitrogen, Grand Island, NY, USA) at 1:5000 dilution with 5 mg/ml RNase A in PBT (PBS+0.1% Tween-20) for 30 min at room temperature. Cy3- or Cy5-conjugated secondary antibodies were used at 1:1000 dilution (Jackson ImmunoResearch, West Grove, PA, USA) and anti-rabbit Alexa Fluor 488 at 1:1000 dilution (Molecular Probes, Grand Island, NY, USA).

## Generation of constructs and cloning

The *kuk-LacZ* construct was synthesized using the 5′UTR and intron of the kuk small transcript (kuk-RB, Flybase). The kuk ORF was replaced by the LacZ coding sequence and was followed by the 3′ UTR of the original transcript (*Figure 5A*). The *kuk-LacZ* construct was fused to a UASg promoter (Gateway system, Invitrogen, Grand Island, NY, USA).

The *4x intron kuk-LacZ* construct was synthesized using 4 repeats of the fragment of 5′UTR and intron of the kuk small transcript, separated by 201 nucleotides of in-frame LacZ sequence. The stop codon is followed by the 3′UTR kuk small transcript sequence and 300 bp of the 3′-located genomic region to promote transcriptional termination (*Figure 6A,D*). In the case of the *no intron kuk-LacZ*, all splice sites (meaning 5′ splice site, branch point, and 3′ splice site) were mutated to thymidines (*Figure 6A,D*). To induce expression of these constructs, they were fused to UAS promoter or *nullo* minimal promoter. The *4x intron kuk-LacZ* and *no intron kuk-LacZ* constructs were cloned into pWALIUM22.

Fandango open reading frame, *kuk-LacZ*, *4x intron kuk-LacZ,* and *no intron kuk-LacZ* constructs were synthetized (GenScript, Piscataway, NJ, USA). Prp19 open reading frame was cloned into pDONR221 from DGC gold BDGP clone LD09231.

Prp19 and Fandango ORFs were subcloned into a vector containing the UASp promoter and 6x C-terminal Myc-tag (Gateway, Invitrogen, Grand Island, NY, USA). All constructs were then used to generate transgenic flies stocks (BestGene, Chino Hills, CA, USA).

## RT-PCR

Total RNA was extracted from 0–3 hr (after egg laying) embryos, unfertilized embryos, and 4-day-old female ovaries mutant for *fandango* and control (FRT42B) following standard procedures (PureLink RNA Mini Kit, Ambion, Grand Island, NY, USA). 1 µg of RNA was then used for reverse transcription with Oligo(dT)12–18 and/or random hexamers primers following the manufacturer's protocol (Transcriptor First Strand cDNA Synthesis Kit, ROCHE, Germany). Primer combinations used were designed with PrimerSelect (Lasergene, Madison, WI, USA) and PCR was performed using GoTaq DNA polymerase (Promega, Fitchburg, WI, USA). Sequences of all primers used are listed in *Supplementary file 2*.

## Real-time qPCR

To measure transcription levels, embryos were staged in three different groups based on the embryonic morphology: stage A (embryos from cycle 1 to 8, no pole cells, and no cortical nuclei are observed); stage B (embryos from cycle 8/9 to 13, pole cells present, and cortical nuclei are observed); and stage C (embryos at interphase 14, blastoderm cellularized). Three independent replicas for each stage, containing each 10 manually selected embryos were generated. Three different genetic backgrounds were analyzed (control (FRT42B), FRT42B *fandango*, and *grapes* as positive control). To measure fandango mRNA levels, unfertilized eggs were analyzed (three replicas). To measure transcription level of the *4x intron kuk-LacZ* and *no intron kuk-LacZ* constructs, 0–3 hr (after egg laying) embryo collections were used to analyze both maternal and zygotic induced expression.

Total RNA was extracted from samples and then used for reverse transcription with Oligo(dT)12–18 as described above. Real-time mRNA quantification was performed following the manufacturer's protocol (QuantiFast SYBR Green RT-PCR Kit, Qiagen, Germany). For analysis of transcription levels of early zygotic genes (*nullo, snail, scute, even-skipped,* and *tailless*) the *Drosophila* QuantiTect Primer Assay (Qiagen, Germany) was used. For mRNA level measurements of fandango, *4x intron kuk-LacZ* and *no intron kuk-LacZ* constructs primers were designed with Primer3 (*Supplementary file 2*).

## Antibodies generated

Anti-Fandango and anti-Prp19 rabbit polyclonal antibodies were raised against recombinant proteins corresponding to amino acids 551–750 of Fandango/CG6197-PA, and to amino acids 20–219 of Prp19-PA, respectively (Metabion international AG, Germany). In both cases it was used His-tagged recombinant proteins as antigen and the antibodies were affinity purified.

## Biochemistry

Protein extracts were obtained from 0–3 hr (after egg laying) embryos or 4-day-old female ovaries. Embryos were dechorionated with 50% commercial bleach solution and ovaries dissected in PBS, samples then homogenized in NB buffer (150 mM NaCl, 50 mM Tris–HCl pH 7,5, 2 mM EDTA, 0,1% NP-40, 1 mM DTT, 10 mM NaF, and EDTA-free protease inhibitor cocktail, Roche, Germany), and centrifuge at 20000×$g$ for 3 min. Supernatant was recovered and centrifuged twice.

To analyze NTC/Prp19 complex composition (*Table 1*), co-immunoprecipitation was done using protein extracts from embryo or ovary tissues expressing Myc-tagged Fandango or Prp19. Briefly, 1 mg of protein was diluted in 1 ml NB buffer and incubated with 1 µg/ml of mouse c-Myc antiboby (9E10) (Santa Cruz Biotechnology, Dallas, Texas, USA) for 1 hr at 4°C. Subsequently, 0.9 mg of Dynabeads Protein G (Invitrogen, Grand Island, NY, USA) were added to the immune complex and incubated 1 hr at 4°C. Beads were washed three times with NB buffer and protein elution performed with 50 µl of 100 mM Glycine pH 2.5 during 2 min at RT and stopped with 5 µl of 1M Tris–HCl pH 10.85. Eluted proteins were then precipitated in five times the volume of acetone at −20°C and samples analyzed by liquid chromatography coupled to tandem mass spectrometry (Mass Spectrometry Laboratory, Institute of Biochemistry and Biophysics, Poland).

To analyze NTC/Prp19 complex composition (showed in *Figure 3A*), protein co-immunoprecipitation was performed using nuclear protein extracts (adapted from *Kamakaka and Kadonaga, 1994*) from a collection of 0–3 hr (after egg laying) wild-type embryos (Oregon-R). 1 mg of protein extract was incubated with rabbit anti-Prp19 (1:1000 dilution) or the pre-immune (1:10,000 dilution) as control, in HNEB2 buffer (100 mM NaCl, 2,5 mM MgCl$_2$, 10 mM Tris–HCl pH 7,5, 0,5% Triton X-100, and EDTA-free protease inhibitor cocktail, Roche, Germany) during 1 hr at 4°C. The procedure was carried out as described in previous co-immunoprecipitation and eluted complexes were boiled in Laemmli sample buffer and analyzed by Western Blot.

Size-exclusion chromatography was performed in protein extracts of 0–3 hr (after egg laying) embryo collections from FRT42B *fand¹* mutants or control (FRT42B). Extracts were prepared as described before in NB2 buffer (150 mM NaCl, 50 mM Tris–HCl pH 7,5, 2 mM EDTA, 0,01% NP-40, 1 mM DTT, and EDTA-free protease inhibitor cocktail, Roche, Germany). Subsequently, 2 mg of protein extract were fractionated using Superose 6 10/300 GL column (GE Healthcare, United Kingdom) in NB2 buffer and fractions collected and analyzed by Western blot.

To analyze protein amount in ovaries and embryos (showed in *Figures 1I and 3C*), embryos were dechorionated and ovaries dissected as described above. Samples were homogenized in PBS supplemented with EDTA-free protease inhibitor cocktail (Roche, Germany) and centrifuged at 20000×$g$ for 3 min at 4°C. Supernatant was collected and protein concentration determined using the Bradford method (BioRad, Hercules, CA, USA). Samples were immediately boiled in Laemmli sample buffer and 10 µg of protein was run in SDS-PAGE gel and analyzed by immunoblot.

Levels of RNApol II CTD Ser2 phosphorylation were analyzed in embryos dechorionated and manually selected at specific developmental stages based on the embryonic morphology (as described above). 15 embryos were selected for each stage and protein sample was obtained by lysing the embryos with a needle in Laemmli sample buffer and heating for 5 min at 100°C. Protein amounts corresponding to ~7 embryos were running in SDS-PAGE and analyzed by immunoblot. Five independent replicas were analyzed.

Antibodies used were: polyclonal rabbit anti-Prp19 at 1:8000 dilution; polyclonal rabbit anti-Fandango at 1:1000 dilution; mouse anti-alpha-Tubulin Dm1A at 1:50,000 dilution (Sigma, St Louis, MO, USA); mouse anti-RNA Polymerase II H5 at 1:500 dilution (MMS-129R, Covance, Princeton, NJ, USA); rabbit anti-ISY1 at 1:500 dilution (ab121250; Abcam, United Kingdom); and mouse anti-CDC5L [2136C1a] at 1:200 dilution (ab51320; Abcam, United Kingdom).

## Mass spectrometry

Peptides mixtures were analyzed by LC-MS-MS/MS (liquid chromatography coupled to tandem mass spectrometry) using Nano-Acquity (Waters, Milford, MA, USA) LC system and Orbitrap Velos mass

spectrometer (Thermo Electron Corp., San Jose, CA, USA). Prior to analysis, proteins were subjected to standard 'in-solution digestion' procedure, during which proteins were reduced with 100 mM DTT (for 30 min at 56°C), alkylated with 0,5 M iodoacetamide (45 min in darkroom at room temperature), and digested overnight with trypsin (sequencing Grade Modified Trypsin—Promega V5111). The peptide mixture was applied to an RP-18 precolumn (nanoACQUITY Symmetry C18—Waters 186003514) using water containing 0,1% TFA as mobile phase, then transferred to nano-HPLC RP-18 column (nanoACQUITY BEH C18–Waters 186003545) using an acetonitrile gradient (0%–35% AcN in 180 min) in the presence of 0.05% formic acid with a flow rate of 250 nl/min. The column outlet was directly coupled to the ion source of the spectrometer, operating in the regime of data dependent MS to MS/MS switch. A blank run ensuring no cross contamination from previous samples preceded each analysis.

Raw data were processed by Mascot Distiller followed by Mascot Search (Matrix Science, London, UK, on-site license) against Flybase database. Search parameters for precursor and product ions mass tolerance were 100 ppm and 0.6 Da, respectively, enzyme specificity: trypsin, missed cleavage sites allowed: 0, fixed modification of cysteine by carbamidomethylation, and variable modification of methionine oxidation. Peptides with Mascot Score exceeding the threshold value corresponding to <5% False Positive Rate, calculated by Mascot procedure, and with the Mascot score above 30 were considered to be positively identified.

Human orthologs were determined using DSRC Integrative Ortholog Prediction Tool (DIOPT) (http://www.flyrnai.org/cgi-bin/DRSC_orthologs.pl). Only scores above two were considered such as the best matches when there was more than one match per input.

## High-throughput transcriptome sequencing (RNAseq)

Total RNA was isolated from 0–3 hr collections of *fandango* maternal mutant and control (FRT42B) embryos using TRIzol Reagent (Invitrogen, Grand Island, NY, USA), following standard protocol. DNAse I (Promega, Fitchburg, WI, USA) treatment was performed during 30 min at 37°C. DNAse was extracted by Phenol-Chloroform extraction; the RNA was precipitated with ethanol, and dissolved it in 25 µl of DEPC water. Bioanalyzer testing was used to analyze quality and concentration of the samples and made up to the volume to 100 µl with water, 50 µl of 7.5 M NH4OAc added, 0.5 µl of glycogen, and 250 µl of absolute ethanol. cDNA library was generated applying the standard Illumina protocol for RNA-Seq (polyA RNAs) and sequenced with an Illumina HiSeq (Oklahoma Medical Research Foundation, Oklahoma City, OK, USA). These generated RNA-Seq data for two biological replicates each of wild-type and *fandango* mutant (*fand¹*), consisting of about 150 million illumina paired-end 100 bp reads per sample.

Paired-end reads were mapped with tophat (*Trapnell et al., 2009*) version 2.0.3 against the *Drosophila melanogaster* BDGP5 reference genome, using Flybase gene annotations downloaded from Ensembl e66 as guide.

To analyze splicing defects, we first extracted exon–intron boundaries from gene annotations. To avoid potential confounding effects, we removed all boundaries that had overlapping exon sequence (from different genes or transcripts). Subsequent analysis used this set of 'safe' exon–intron boundaries.

For coverage plots in *Figure 2D*, we also excluded boundaries where the intron or the exon were less than 50 b long. At each base within 50 bp either side of a splice site we count the number of reads that overlap that base, then divide by the total number of reads within the 100 bp centered around the splice site. To minimize noise, we require that at least 50 reads fall within the −50:50 window around the exon–intron boundary (reads that only partially overlap the window are also counted).

To determine the frequency of splicing defects for each boundary, we extracted all reads that overlap the 5' splice site by at least 10 bp to either side of the boundary. We classified each read as correctly spliced (if the read is split from the 5' to the 3' splice site), unspliced (if the read is not split) or mis-spliced (if the read is split but not matching the expected 5' or/and 3' sites). To reduce noise, we only include an exon–intron boundary if at least 10 fragments overlap that boundary.

To determine the exon–intron gene structure (*Figure 2—figure supplement 2D*), each gene was divided in 1000 equal segments. For each segment of each gene, we checked for the presence (or absence) of an exon in that segment. For each segment, we then plotted the frequency of exon presence in all genes. If an exon randomly appears in a given segment, it appears in ~50% of genes. For example, a set of intronless genes would produce a plot that would be always at 100%.

To determine the splice site motif (*Figure 2—figure supplement 2C*), sequences around exon–intron boundaries were extracted and motifs drawn using Weblogo.

Early zygotic and maternal genes were defined using RNA-Seq developmental gene expression data from Flybase (*Graveley et al., 2011*). A gene was defined as an early zygotic gene when its expression at 2–4 hr is at least moderate (more than 10 expression units, as defined in the Flybase dataset) and at least 5× greater than its expression at 0–2 hr (irrespective of the 0–2 hr value). Maternal genes are those genes that are not early zygotic and have high expression (more than 50 expression units) at 0–2 hr. To avoid potential artifacts, genes that have an extremely high expression (more than 1000 expression units) were not considered. Applying this definition we obtained 270 early zygotic genes (including 43 genes from *De Renzis et al., 2007*) and 2048 maternal genes.

All scripts used for this analysis are available upon request. RNA-Seq data are available in the ArrayExpress database (www.ebi.ac.uk/arrayexpress) under accession number E-MTAB-2321.

### In situ hybridization

The procedure has been described in *Stein et al. (2002)*. Antisense digoxigenin-labeled RNA probes were synthesized using the DIG RNA labeling Kit (Roche, Germany). *eve* and *nos* probes were made from pBluescript plasmids containing the respective cDNAs.

### Sequence alignment

Sequences were aligned using ClustalW2 (http://www.ebi.ac.uk/Tools/msa/clustalw2/) and BoxShade 3.21 (http://www.ch.embnet.org/software/BOX_form.html) for printing and shading of multiple alignment file.

### Statistical analysis

Unpaired $t$ test and two-way ANOVA were performed using Prism 5.00 for Windows (GraphPad Software, San Diego, CA, USA).

## Acknowledgements

Both *fandango* mutant alleles were isolated in the laboratory of Ruth Lehmann. We thank Kohtaro Tanaka for help with the *in situs*; our colleagues, Moises Mallo, Miguel Ferreira for discussion, and suggestions that greatly improved the manuscript; Jessica Thompson and Richard Hampson for manuscript editing. Proteins were identified at the Mass Spectrometry Laboratory Institute of Biochemistry and Biophysics Polish Academy of Science.

## Additional information

### Funding

| Funder | Grant reference number | Author |
| --- | --- | --- |
| FCT-Fundacao para a Ciencia e Tecnologia (Portugal) | PTDC/SAU-BID/111796/2009 | Rui Gonçalo Martinho |
| FCT-Fundacao para a Ciencia e Tecnologia (Portugal) | PTDC/BIA-BCM/111822/ 2009 | Rui Gonçalo Martinho |
| FCT-Fundacao para a Ciencia e Tecnologia (Portugal) | PTDC/BBB-BQB/0712/2012 | Rui Gonçalo Martinho |
| FCT-Fundacao para a Ciencia e Tecnologia (Portugal) | PEst-OE/EQB/LA0023/2013 | Rui Gonçalo Martinho |
| FCT-Fundacao para a Ciencia e Tecnologia (Portugal) | SFRH/BPD/47957/2008 | Leonardo Gastón Guilgur |
| FCT-Fundacao para a Ciencia e Tecnologia (Portugal) | SFRH/BPD/63869/2009 | Denisa Liszekova |

The funders had no role in study design, data collection and interpretation, or the decision to submit the work for publication.

## Author contributions

LGG, Conception and design, Acquisition of data, Analysis and interpretation of data, Drafting or revising the article; PP, Conception and design, Acquisition of data, Analysis and interpretation of data; DS, DL, AR, Acquisition of data, Analysis and interpretation of data; RGM, Conception and design, Analysis and interpretation of data, Drafting or revising the article

## Author ORCIDs

Rui Gonçalo Martinho, [iD] http://orcid.org/0000-0002-1641-3403

## Additional files

### Supplementary files

• Supplementary file 1. Complete list of proteins specifically co-immunoprecipitating with Myc-tagged Fandango and Myc-tagged Prp19. List of proteins co-immunoprecipitating with Myc-tagged Fandango in ovaries (germ-line) and embryos, and with Myc-tagged Prp19 in embryos. Protein extracts from embryos and ovaries not expressing the Myc-tagged proteins were used as negative controls (−). Two replica experiments were performed for each condition. Proteins were identified by LC-MS-MS and data were blasted against the Flybase database using Mascot Search (see 'Materials and methods' for detail). The presence of a protein was measured from the total number of peptides (Matches), the number of non-repeated sequence peptides (Sequences) and the respective Score according to Mascot Search. Predictions of Human protein orthologous from *Drosophila* proteins were made using DIOPT (see 'Materials and methods' for detail). Proteins were grouped in NTC/Prp19-related complex (light gray), NTC/Prp19 complex (dark gray), general spliceosomal related proteins, miscellaneous, and ribosomal proteins. Non-reproducible (only 1 peptide in one replica) and non-specific proteins (more than 1 peptide in negative controls [−]) are listed separately. (*) Spliceosomal proteins in non-reproducible table, (**) NTC/Prp19 complex or NTC/Prp19-related complex subunits present in non-reproducible and non-specific tables.

• Supplementary file 2. Complete list of primers. Sequences of all the primers used in the RT-PCR and real-time qPCR assays presented in this manuscript ('Materials and methods').

### Major dataset

The following dataset was generated:

| Author(s) | Year | Dataset title | Dataset ID and/or URL | Database, license, and accessibility information |
|---|---|---|---|---|
| Guilgur Leonardo, Prudêncio Pedro, Sobral Daniel, Liszekova Denisa, Rosa André, Martinho Rui | 2014 | RNA-seq of coding RNA of fandango, a Drosophila melanogaster mutant affecting splicing | http://www.ebi.ac.uk/arrayexpress/experiments/E-MTAB-2321/ | Publicly available at EMBL-EBI. |

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
