## [Decision Letter]

Thank you for sending your work entitled “Requirement for highly efficient pre-mRNA splicing during Drosophila early embryonic development” for consideration at *eLife*. Your article has been favorably evaluated by a Senior editor and 2 reviewers, one of whom, Elisa Izaurralde, is a member of our Board of Reviewing Editors.

The Reviewing editor and the other reviewer discussed their comments before we reached this decision, and the Reviewing editor has assembled the following comments to help you prepare a revised submission.

The manuscript by Guilgur et al. identifies and characterizes mutations in Drosophila genes encoding components of the spliceosomal NTC/Prp19 complexes, a critical component of spliceosome activation. Interestingly, the authors describe a differential requirement for the NTC/Prp19 components in the splicing of transcripts from maternally versus early zygotically expressed genes in the Drosophila embryo. The pre-mRNA splicing defects detected in these mutants correlate with zygotic gene expression and the rapid mitoses characteristic of cell division in early Drosophila embryos. These observations are interesting in light of two related findings. First, that mitosis inhibits pre-mRNA splicing in organisms that undergo nuclear membrane breakdown. Second, in yeast intron size is small and most genes lack introns suggesting that the genomes of rapidly dividing cells were selected for intron loss.

Additionally, the manuscript reveals a feedback or coordination between the time required to complete nuclear division and architecture of expressed genes. The mechanism remains unclear, but this study opens new research avenues. For example, what happens with pre-mRNAs that have “no time” to be spliced. Do they accumulate in the nucleus for later splicing or are they degraded by nuclear quality control (NQC) mechanisms. Another interesting question is how splicing efficiency adapts to cell cycle duration, and how a given transcript is processed in highly and non-proliferating cells.

Although there are many questions that remain unanswered, the finding that there is an evolutionary pressure for early zygotic genes to lack introns is novel. Additionally, the proposed coordination between the time to complete nuclear division and splicing efficiency provides an interesting mechanism to regulate gene expression and forms the basis for future studies.

Overall, the paper is interesting, the data and observations support the conclusions drawn and should be published with minor revisions, outlined below.

1) There is almost too much data in this paper, including a lot in the supplemental materials. Even the main figures often have multiple panels, some of which could be moved to the supplemental materials to simplify presentation of the main and seminal points.

2) There are numerous grammatical or typographical errors that need to be fixed before publication.

3) Finally, although the authors imply this, they should state explicitly the connections between the small or no intron containing zygotic genes, rapid cell divisions in the early embryo and intron loss in yeast and single cell eukaryotes.

---

## [Author Response]

*1) There is almost too much data in this paper, including a lot in the supplemental materials. Even the main figures often have multiple panels, some of which could be moved to the supplemental materials to simplify presentation of the main and seminal points*.

We agree with the reviewers. In order to simplify the presentation of the main points of the manuscript and increase its sharpness, the manuscript now has only six figures, instead of the original seven. Although none of the originally presented results were excluded, some were moved (according to reviewers suggestion) to Figure supplements.

More specifically, a) “Figure 3” is now “Figure 2—figure supplement 2”. b) Panel B of “Figure 5” (now: Figure 4), was moved to “Figure 4—figure supplement 1”

*2) There are numerous grammatical or typographical errors that need to be fixed before publication*.

With the help of an English native speaker we have extensively edited the manuscript. This editing has in no way altered the scientific content of this manuscript and its main conclusions.

*3) Finally, although the authors imply this, they should state explicitly the connections between the small or no intron containing zygotic genes, rapid cell divisions in the early embryo and intron loss in yeast and single cell eukaryotes*.

We include a new paragraph and additional citation in the Discussion where we suggest that, similarly to the early embryo, rapid cell divisions can also potentially act as driving force for changes in gene architecture in unicellular organisms:

“Our results highlight cell cycle constraints during early embryo development as a force capable of driving changes in gene architecture of multicellular organisms. In unicellular organisms intron paucity correlates with a bias toward the 5’ ends, whereas introns from multicellular genomes are evenly distributed throughout the genes (29). This suggests that similar constraints on gene architecture are also likely to exist in yeast and other fast-dividing single cell eukaryotes.”